# Study on the Fabrication of Porous TiAl Alloy via Non-Aqueous Gel Casting of a TiH$_2$ and Al Powder Mixture

**Fei Li** [1,2,*] **, Xiao Zhang** [3,*]**, Yi Jiang** [1]**, Lixiang Yang** [1]**, Chengkang Qi** [1,*] **and Baode Sun** [2]

[1]  Shanghai Key Laboratory for High Temperature Materials and Precision Forming, Shanghai Jiao Tong University, Shanghai 200240, China; MarzPicture@sjtu.edu.cn (Y.J.); yanglixiang@sjtu.edu.cn (L.Y.)
[2]  State Key Laboratory of Metal Matrix Composites, Shanghai Jiao Tong University, Shanghai 200240, China; bdsun@sjtu.edu.cn
[3]  State Key Laboratory of Special Rare Metal Materials, Northwest Rare Metal Materials Research Institute Ningxia Co. Ltd., Shizuishan 735000, China
[*]  Correspondence: lifei74@sjtu.edu.cn (F.L.); guofangkeji@163.com (X.Z.); chengkangq@sjtu.edu.cn (C.Q.); Tel.: +86-21-3420-2951 (F.L.); +86-952-2099-011 (X.Z.); +86-21-5474-2661 (C.Q.)

**Abstract:** A porous TiAl alloy with 23.78% porosity was successfully fabricated via a low-toxicity, non-aqueous gel casting method by using a titanium hydride (TiH$_2$) and aluminum (Al) powder mixture as the raw material. The effects of dispersant content and solid loading on the rheological properties of the TiH$_2$/Al slurries were studied systematically. It was found that all the slurries exhibited a typical shear-thinning behavior, which is favorable for the gel casting process. Three-point bending tests of the dried TiH$_2$/Al green bodies were carried out, and the results showed that the flexural strength was raised from 28.86 to 62.36 MPa with increasing monomer (hydroxyethyl methacrylate, HEMA) content. In order to study the degreasing process and minimize the possible residual carbon and oxygen after sintering, TGA analysis was performed. The fracture morphology of the sintered TiAl alloy (1400 °C for 2 h) was studied by scanning electron microscope (SEM). Based on the X-ray diffraction (XRD) identification, the main phases of the sintered part were γ-TiAl, α$_2$-Ti$_3$Al, and a small amount of Al$_2$Ti and Al$_3$Ti.

**Keywords:** TiAl alloy; gel casting; titanium hydride; sintering; porosity

## 1. Introduction

On the basis of their low density, high modulus, high temperature strength, good flame retardancy, excellent oxidation resistance, and creep resistance, TiAl alloys have been developed into one of the most attractive materials for lightweight and heat-resistant structural parts, such as those used in aviation, aerospace, aeronautical missiles, automotive engines, and so on [1–5]. In recent years, there has also been great interest in porous TiAl alloys as potential engineering materials for various industrial fields [6–9]. However, the traditional processing routes for preparing TiAl alloys, such as ingot metallurgy, hot forging, precision casting, powder metallurgy, and machining [10–14], either have long production cycles or waste quite a few materials, leading not only to high manufacturing costs, but also to difficulty in fully realizing the advantages in the preparation of parts with complex shapes. Therefore, it is of great importance to apply an appropriate technology to overcome the shortcomings mentioned above.

Gel casting, as a near net shaping technology, was firstly developed by Janney and Omatete in the early 1990s and has been widely applied in the ceramic industry [15–18]. In such a process, a high-solid-loading slurry consisting of raw powders, solvent, and organic binder is first required.

After the slurry is casted into a mold, the organic binder creates a macromolecular network to hold the ceramic particles together. Homogeneous green bodies with the required mechanical properties are therefore obtained. In the past few decades, a considerable amount of research has been done on the development of the ceramic gel casting technique. Along with the constant maturity of gel casting technology, the applications have been expanded to the domain of metal materials [19–22].

It is well known that the performance of titanium-based objects can be easily deteriorated by impurities resulting from residual oxygen, nitrogen, and carbon during manufacturing [23]. Due to the high stability of these impurities, such as $TiO_2$ and TiN, it is impossible to eliminate them by using titanium or its elemental powder as a raw material. Fortunately, titanium hydride can be an ideal raw material to reduce the content of these impurities. In a previous work, the powder metallurgy of $TiH_2$ and $TiH_2$/Al was studied. Researchers, such as Kendra and Guo, also carried out impressive works on the gel casting of $TiH_2$, and their results showed that oxygen and carbon contamination could be reduced [24–26].

In this study, a porous TiAl alloy was fabricated via non-aqueous gel casting followed by a vacuum sintering process by using $TiH_2$/Al as raw powders. The main purpose of the present work is to further develop a simple approach for TiAl structural material production which could meet the requirements of reducing the cost and expanding application areas.

## 2. Materials and Methods

The low-toxicity gel casting system used was composed of N, N-dimethylformamide (DMF) as solvent, hydroxyethyl methacrylate (HEMA) as monomer, 1,6-hexanediol diacrylate (HDDA) as cross-linker, tert-butyl peroxybenzoate (TBPB) as initiator, and polyvinyl pyrrolidone (PVP) as dispersant. All the reagents were chemically pure and supplied by Shanghai National Pharmaceutical Reagents Group Co. Ltd., China. Commercially available 99.7% pure $TiH_2$ powder (supplied by Shanghai Bike New Materials Co. Ltd., Shanghai, China) and 99.85% pure Al powder (supplied by Shanghai Bike New Materials Co. Ltd., Shanghai, China) were used as raw materials.

Premixed solutions containing 15 vol % to 40 vol % monomer relative to the volume of the solutions were prepared by dissolving HEMA, HDDA, and PVP in DMF at room temperature by means of ultrasonic dispersion. A $TiH_2$ and Al powder mixture with a molar ratio of 1:1 was added to the solutions and mixed by a stirrer for about 30 min to obtain 40 vol % to 48 vol % $TiH_2$/Al slurries. In all experiments, the ratio of HEMA to HDDA was fixed at 15:1. Later, a little dose of TBPB was dropped into the slurries and fully stirred. After that, the slurries were poured into individual silicone molds and held at 80 °C in a vacuum drying chamber for 2 h to complete the gelation process. Subsequently, the wet green bodies were carefully removed from the molds and soaked in ethanol for 8 h in order to remove residual solvent. To avoid cracking during drying, the green bodies were dried in a vacuum chamber at 25, 80, and 120 °C for 8 h in sequence. Finally, the $TiH_2$/Al green bodies were sintered at 1400 °C under vacuum conditions for 2 h.

The particle size distributions of the $TiH_2$ and Al powders were measured by a laser analyzer (S3500, Microtrac, USA). The micromorphology of the powders and the fracture surfaces of the green and sintered bodies were observed by a scanning electron microscope (Nova Nano SEM 230, FEI, USA). The rheological behavior of the $TiH_2$/Al slurries was measured at room temperature by a rotary rheometer (Gemini 200HR, Bolin, UK) with the shearing rate ranging from 0 to 1000 s$^{-1}$. The flexural strength of the green bodies was measured by a universal testing machine (Z020, Zwick/Roell, German) with a crosshead speed of 1 mm/min. Samples with a size of 40 mm × 5 mm × 5 mm for the test were held in a vacuum chamber at 120 °C for 2 h to avoid the influence of moisture before testing. The degreasing process of the dried green body was studied by using a TG analyzer (TGA8000TM, PerkinElmer, USA), and the samples were heated at a rate of 10 °C /min from room temperature to 700 °C under a pure argon flow. An X-ray diffraction analyzer (D/max 2500, Rigaku, Japan) was used to identify the phases of the sintered samples. The porosity and pore size distribution of the sintered component were detected by a mercury intrusion method.

## 3. Results

The SEM micrographs of the TiH$_2$ and Al powders are shown in Figure 1a,b, respectively. It can be seen from Figure 1a that most TiH$_2$ particles have an irregular polygonal shape and a rugged surface. Al particles are mostly hollow and spherical and much smaller than TiH$_2$ particles. The particle size distribution analysis is presented in Figure 2, and the result shows that the TiH$_2$ particles have a wide size distribution with a D50 of 21.7 μm, while the Al particles have a narrow size distribution with a D50 of 1.58 μm. The results of the size distributions are in keeping with the SEM micrographs in Figure 1.

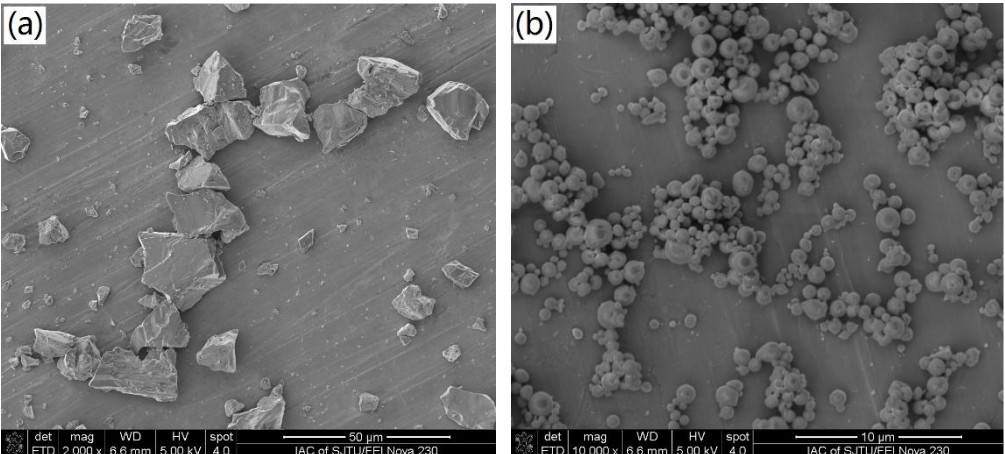

**Figure 1.** SEM micrographs of (**a**) TiH$_2$ and (**b**) Al particles.

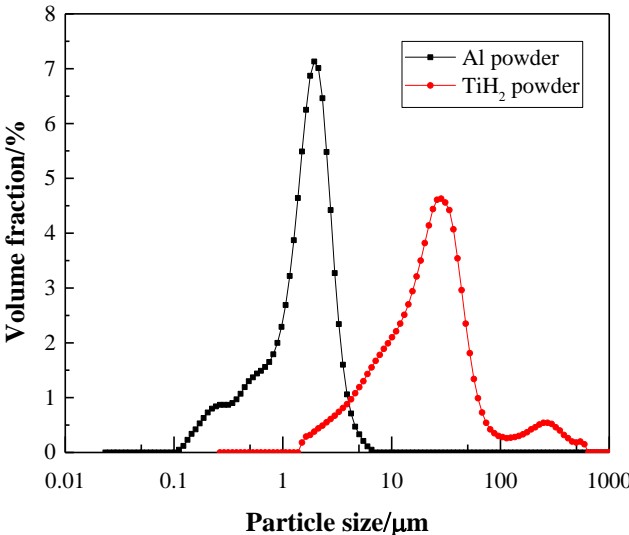

**Figure 2.** Size distribution curves of TiH$_2$ and Al Powders.

For gel casting, it is quite essential to obtain stable slurries with good fluidity; hence, dispersants are added to stabilize solid particles and avoid aggregation or subsidence. There are mainly three mechanisms of dispersion, namely, steric hindrance, electrostatic repulsion, and their combination. In non-aqueous gel casting, steric hindrance dominates [27]. Therefore, polymer dispersants which have a strong steric effect are preferred. In this study, PVP was chosen as the dispersant, and the rheological properties of TiH$_2$/Al slurries with a solid content of 46 vol % were measured by a rotary rheometer. During the preparation of TiH$_2$/Al slurries, it was found that when the ratio of PVP to powder mass was less than 1 wt %, TiH$_2$ particles were easily sedimented due to their big size and

density of 3.8 g/cm$^3$, which would result in a non-uniform distribution of powders. Hence, a sufficient dose of PVP was needed. Three PVP addition ratios relative to the TiH$_2$/Al powder mixture were chosen: 1, 2, and 3 wt %. The effects of the dispersant dosage on the rheological behavior of the TiH$_2$/Al slurries are shown in Figure 3. It can be seen that all the TiH$_2$/Al slurries exhibited a shear-thinning behavior over the measured shear rate range, which indicates that the TiH$_2$ and Al particles were in the flocculation state in the static slurries. When shear force was applied, the flocculation structure was dismantled and the apparent viscosities decreased. Once the flocculation structure in the slurries was completely dismantled, the viscosities did not decrease anymore. In addition, both the apparent viscosity and the shear stress of the slurries increased while the PVP content rose from 1 to 3 wt %. At the shear rate of 100 s$^{-1}$, the slurry prepared from the precursor solution containing 1 wt % PVP relative to the TiH$_2$/Al powders exhibited a shear viscosity of 169.3 mPa·s, while the slurries with 2 wt % and 3 wt % PVP displayed viscosities as high as 382.6 mPa·s and 919.2 mPa·s, respectively. For the sake of adequate mold filling, TiH$_2$/Al slurries with low viscosity were necessary, and 1 wt % was chosen as the optimum amount of PVP addition to the slurry.

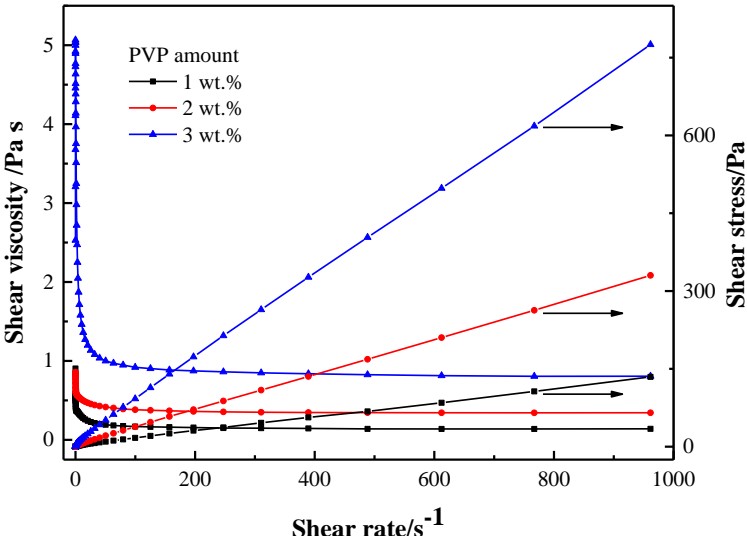

**Figure 3.** Effects of the PVP dispersant amount on the rheological properties of the TiH$_2$/Al slurries.

For the gel casting process, the solid loading of the slurry has great effects on the fluidity of the slurry and the density of the green and sintered parts. Low viscosity of the slurry is conducive to uniform filling of the mold, while high solid loading has the benefit of reducing the volume shrinkage and increasing the sintering density of the green body during drying and sintering, which is particularly important for the deformation control of products with a complex structure. However, low viscosity and high solid loading of the slurry often contradict each other. Therefore, a balance between the two must be considered when preparing a gel casting slurry.

Figure 4 shows the respective rheological curves of the TiH$_2$/Al slurries with solid loadings of 40, 42, 44, 46, and 48 vol %. As expected, the apparent viscosity and shear stress of the slurries displayed a remarkable increasing tendency with the rise of the solid loading content. The slurry with a solid loading of 46 vol % possessed a shear viscosity of 393.5 mPa·s at the shear rate of 100 s$^{-1}$ and an intimal viscosity less than 1000 mPa·s, which is enough to meet the requirements of gel casting. The shear stress versus shear rate in Figure 5 could be further analyzed using the Herschel–Buckley model [28]:

$$\tau = \tau_0 + k\gamma n \tag{1}$$

where $\tau$ is the shear stress, $\tau_0$ the yield stress, k is a consistency coefficient, $\gamma$ is the shear rate, and n is the flow behavior index. According to the rheological properties of the TiH$_2$/Al slurries shown in Figures 3 and 4, it is evident that the slurries belong to typical pseudoplastic fluids [28].

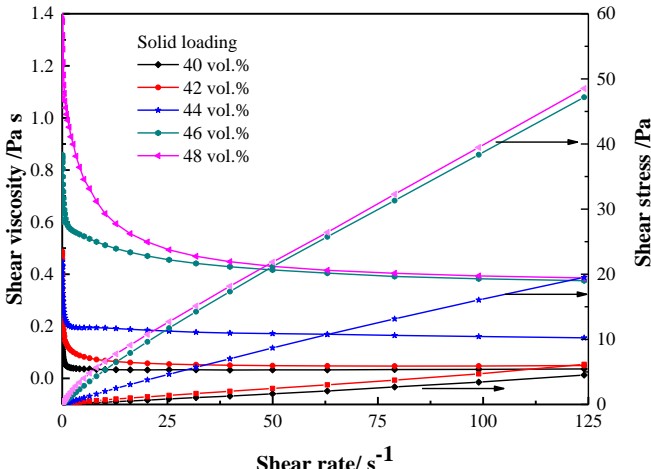

**Figure 4.** Effects of the solid loading on the rheological properties of the TiH$_2$/Al slurries.

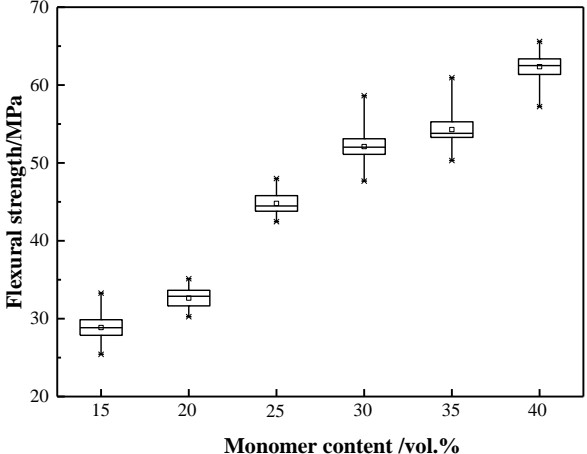

**Figure 5.** Effect of monomer content on flexural strength of dried bodies derived from the 46 vol % TiH$_2$/Al slurry.

In order to avoid cracking during demolding, the strength of the gel-casted body must be considered while fabricating complex-shaped parts. Figure 5 shows the effect of monomer contents on the flexural strength of the green bodies. It can be seen from Figure 5 that with the increase of the monomer contents from 15 to 40 vol %, the flexural strengths of the green bodies increased from 28.86 to 62.36 MPa. This might be due to the increasing monomer content bringing more polymer networks in per unit volume, leading to increased strength.

Figure 6 shows the fracture micrograph of TiH$_2$/Al green bodies with different monomer contents when the solid loading is 46 vol %. It can be seen from Figure 6 that when the gelled polymer contents in the green bodies are relatively low (Figure 6a–d), the spherical state of the Al particles can still be observed, while the TiH$_2$ particles are also encapsulated by the polymer and partly covered by Al particles. Meanwhile, there are some pores in those bodies. Some of the pores could be attributed to the pull-out of the TiH$_2$ particles when the green body was broken off. It was also found from the experiment that when the monomer content is too high (for example, over 35 vol %), the slurry solidifies quickly and is not easy to control. Meanwhile, in the processes of degreasing and sintering, more polymer in the green bodies might lead to more residues, which would have adverse effects on

the properties of the sintered bodies. Therefore, the suitable monomer dosage determined in this study was 30 vol %.

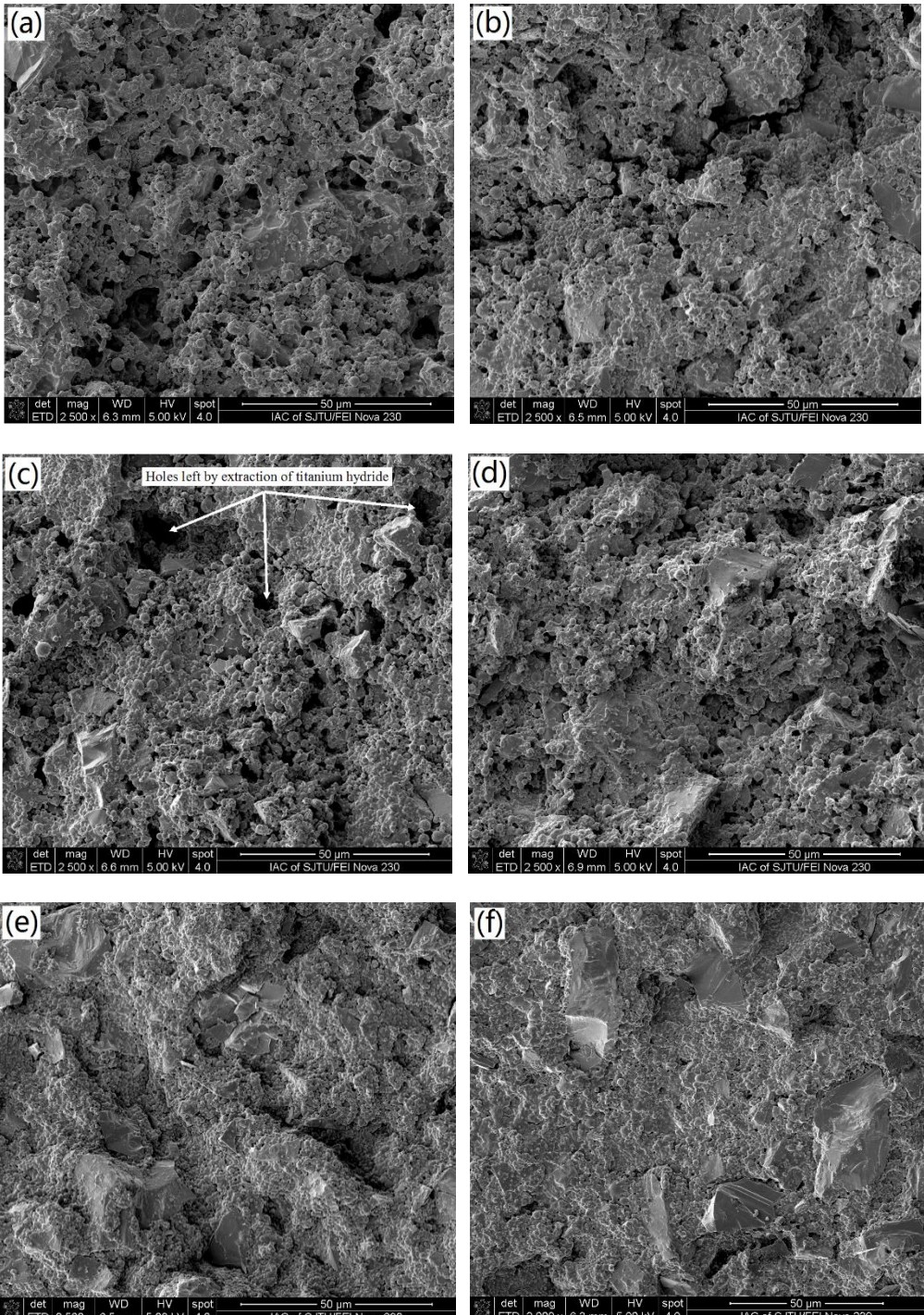

**Figure 6.** Cross-sectional micrographs of TiH$_2$/Al green bodies gel casted from slurries with different monomer contents. (**a**) 15 vol %, (**b**) 20 vol %, (**c**) 25 vol %, (**d**) 30 vol %, (**e**) 35 vol %, and (**f**) 40 vol %.

Figure 7 shows the thermogravimetric curve of the green body with a monomer content of 30 vol % and a solid loading of 46 vol %. As can be seen from Figure 7, the thermogravimetric curve of the TiH$_2$/Al green body can be divided into four stages (s1–s4). The first stage is from room temperature to 300 °C. In this temperature range, the weight loss of the sample is about 2%, which corresponds to the

volatilization of the solvent and the initial decomposition of the gelled polymer. The second stage is from 300 to 500 °C. In this temperature range, the weight loss of the sample is about 9%, corresponding to the massive decomposition of the polymer network, and the DTG curve also indicates that the decomposition rate reaches its maximum value near 418 °C. The third stage is from 500 to 620 °C. The weight of the sample remains unchanged, indicating that the burning of the polymer is basically completed, but $TiH_2$ does not dehydrogenate obviously. In the fourth stage, from 620 to 700 °C, a small amount of thermal weight loss occurs, corresponding to the dehydrogenation process of $TiH_2$, i.e., $TiH_2 \rightarrow TiHx \rightarrow \alpha$-Ti $(0.7 < x < 1.1)$ [29]. According to the above thermal analysis results, $TiH_2$/Al green bodies should be kept at a temperature range of 300 to 500 °C for a long time under a low heating rate during sintering to ensure that all the gelled polymer is decomposed completely.

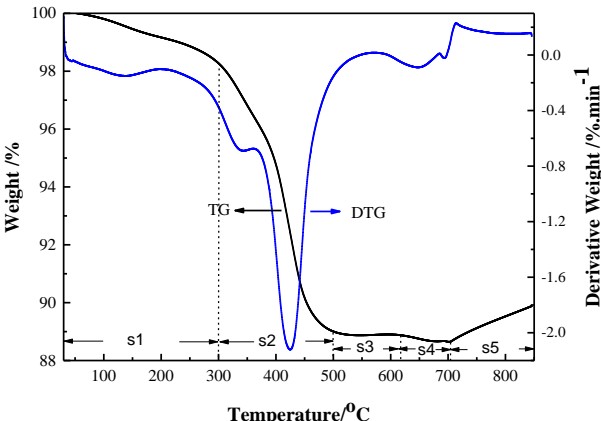

**Figure 7.** Thermogravimetric curve of the $TiH_2$/Al green body by TGA.

Figure 8 shows the microstructure of the TiAl alloy sintered at 1400 °C under vacuum conditions for 2 h. It can be seen from Figure 8 that some open pores with sizes of several microns exist in the sintered part with a typical fracture cleavage surface. The peak intensities from an XRD pattern (Figure 9) of TiAl alloys sintered at 1400 °C mainly identified the phases as $\gamma$-TiAl and $\alpha_2$-$Ti_3$Al with a small amount of $TiAl_2$ and $TiAl_3$. It is evident that residuals from organic additives had been eliminated since no carbide phase was detected by XRD. Figure 10 is the pore size distribution curve for the TiAl alloy sample sintered at 1400 °C. It can be seen from Figure 10 that the sample has a narrow pore size distribution of 2 to 8 microns and its median pore diameter is 3.57 microns, which matches the SEM observation shown in Figure 8. The porosity and apparent density of the as-sintered TiAl alloy are 23.78% and 4.01 $g/cm^3$, respectively.

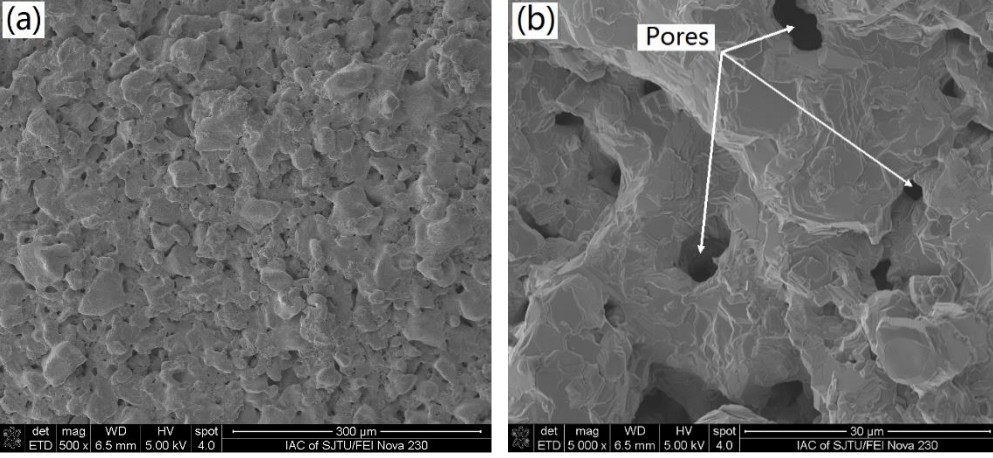

**Figure 8.** SEM micrographs of sintered TiAl alloys. (**a**) 500 × and (**b**) 5000 ×.

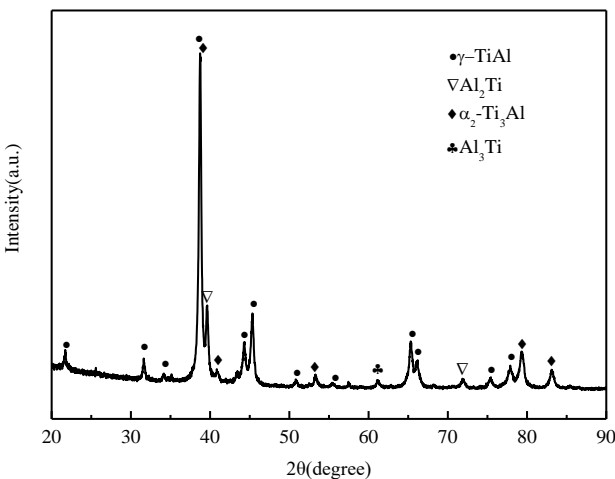

**Figure 9.** XRD pattern of TiAl alloy after the sintering process.

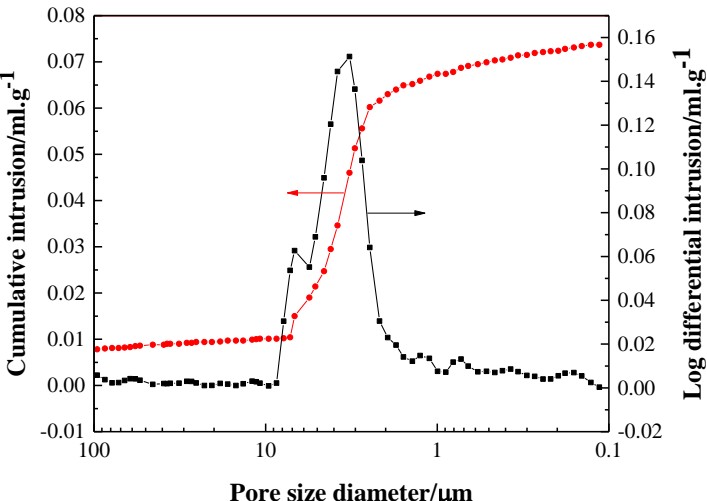

**Figure 10.** Pore size distribution of the TiAl alloy sintered at 1400 °C in vacuum for 2 h.

## 4. Conclusions

This work described a non-aqueous gel casting process with potential for fabricating porous and complicated TiAl components. Based on the characterization of the raw materials, the rheological properties of the slurries for gel casting, the mechanical and thermal properties of green bodies, and the microstructure and phase composition of the sintered part presented above, the following conclusions can be drawn:

1. A non-aqueous $TiH_2$/Al slurry system for gel casting was developed, which exhibited shear-thinning behavior and was favorable for the gel casting process of $TiH_2$/Al green bodies with a complex shape. The optimum amount of PVP addition to the slurry was 1 wt % relative to the $TiH_2$/Al powder mixture.

2. The monomer contents have a great effect on the mechanical properties of the $TiH_2$/Al green bodies. The flexural strength of the green bodies increased from 28.86 to 62.36 MPa as the monomer contents in the premix solution increased from 15 to 40 vol % relative to the premix solution. The suitable monomer dosage determined was 30 vol % relative to the volume of the solutions.

3. The sintered TiAl alloy exhibited a typical fracture cleavage surface; meanwhile, its pore size distribution was 2 μm to 8 μm and its porosity was 23.78%, which indicates that the gel casting route is a suitable way to prepare porous TiAl alloy components. The result of the XRD analysis

shows that γ-TiAl and α₂-Ti₃Al were the main phases and no contaminated carbide phase was formed.

**Author Contributions:** Conceptualization: F.L.; methodology: F.L. and X.Z.; investigation: L.Y., Y.J. and X.Z.; data curation: C.Q.; writing—original draft preparation: F.L.; writing—review and editing: C.Q.; supervision: B.S.

**Funding:** This research was funded by the National Science and Technology Major Project "Aero engine and Gas Turbine," grant number 2017-VII-0008-0102, the Shanghai Municipal Science and Technology Innovation Action Plan under Project, grant number CXY-2016-004 and the "13th Five-Year" equipment pre research and sharing technology project, grant numbers 41423040203 and 41423040206.

**Acknowledgments:** We thank Ying Zhang (Analysis and Testing Center of Shanghai Jiao Tong University) for her help with running SEM pictures. We also gratefully acknowledge helpful discussions with Yanjie Zhao and Junhua Lai.

**Conflicts of Interest:** The authors declare no conflict of interest.

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
