# Peer review of "Study on the Fabrication of Porous TiAl Alloy via Non-Aqueous Gel Casting of a TiH2 and Al Powder Mixture"

_applsci, doi:10.3390/app9081569_

Round 1

Reviewer 1 Report

- According to the SI unit system there must be a space between numbers and units; sometimes the space is there and sometimes not - change that please

- nothing more to complain about your work

Author Response

Dear reviewer,

Thank you for the valuable comments and suggestions on our manuscript. We have modified the manuscript accordingly. The detailed corrections are listed below point by point:

- According to the SI unit system there must be a space between numbers and units; sometimes the space is there and sometimes not - change that please.

Answer: We have added the spaces between numbers and units throughout the manuscript. At the same time, we have also improved the grammar of the manuscript.

The manuscript has been resubmitted to the journal. We are looking forward to your positive response.

Sincerely,

Dr. Fei Li

Shanghai Key Laboratory for High Temperature Materials and Precision Forming, Shanghai Jiao Tong University, Shanghai 200240,China

Email address: lifei74@sjtu.edu.cn

Tel/Fax: +86-21-34202951

Reviewer 2 Report

This paper has shown fabrication of porous TiAl alloys by a gel casting process using TiH2 and Al powders. The detailed gel casting processes, mechanical properties in terms of flexural strength of green body, flow properties in terms of shear viscosity along with vol.% of binder and solid and the microstructural interpretation using SEM and XRD seemed to be well addressed. However, several comments must be taken into accounts in order to meet the high standard of scientific contents as well as good quality of presentation in this journal.

The title itself seems not to be appropriate for the whole stream of this work. It shall be modified; e.g. “ Study on fabricating porous TiAl alloy via non-aqueous gel- casting of TiH2 and Al powder mixture”

English skill seems not to be in good quality throughout the manuscript. Some sentences are barely understood. It must be reviewed and corrected. For example, line 41-44, line 82-86, and so on.

In the the line 75, green bodies were dried at 25, 80 and 120. Was it done in a sequence or done separately? If was this process done separately, the green samples were tested separately?

In the line 99, authors claimed “Due to the irregular morphology and large size of TiH2 particles, it is easy to form "bridges" between TiH2 particles in the green body, while fine Al particles can fill the gap between TiH2 particles and improve the density of the green body.” This sentence was not supported by evidences or relevant references. Otherwise, it shall be rewritten in a speculation manner.  

In the line 118, the sentence, “There are mainly three mechanisms of dispersion, namely, steric hindrance, electrostatic repulsion and their combinations. As for non-aqueous gel-casting, steric hindrance dominates”, needs to have relevant references.

In the line 114, authors claimed “precipitation”. It seems not to be correct expression. It is better to change this expression.

In the line 114, authors claimed “pseudo-plastic fluids”. Proper definition or relevant references shall be provided.

In the line 154, authors claimed “This is because that the increasing of monomer content brings more polymer networks in per unit volume, leading to the growth of strength”. This sentence was not supported by evidences or relevant references. Otherwise, it shall be rewritten in a speculation manner.

In the lines 166-170, authors claimed that bonding strength of polymer network is much stronger than TiH2 particles. In the figure 6 (f), TiH2 particles were fractured during the 3 point bend test. However, it is hardly acceptable. According to the references (J. J. Xu wt al, J. of Alloys and Compound, 436, 2007, p.82, and “Ti powder Metallurgy; Science, Technology and Application”, eds, Ma Qian and F. H. Froes, 2016, p.128, Butterworth-Heineman), yield stress of TiH2 is 774-1037 MPa, and 470-480 MPa, respectively. Under the flexural strength level of this test condition TiH2 powders would be safe. The morphology of TiH2 powders in Fig. 1(a) looks facetted and the morphology which is claimed as “fractured” in Fig 6(f) could not be distinguished with pristine powders clearly. This fact shall be reconsidered carefully.

In the line 198, authors again claimed “The fracture cleavage surface of typical TiAl intermetallics is shown in the section morphology”. The fact was observed after a mechanical testing until fracture?  If so, mechanical testing after sintering shall be discussed in the experimental and discussion.

Author Response

Dear reviewer,

Thank you for the valuable comments and suggestions on our manuscript. We have modified the manuscript accordingly. The detailed corrections are listed below point by point:

1. The title itself seems not to be appropriate for the whole stream of this work. It shall be modified; e.g. “Study on fabricating porous TiAl alloy via non-aqueous gel- casting of TiH2 and Al powder mixture”

Answer: According to the reviewer's suggestion, the title of this paper has been modified as “Study on fabricating porous TiAl alloy via non-aqueous gel casting of a TiH2 and Al powder mixture”. Please see the revised manuscript.

2. English skill seems not to be in good quality throughout the manuscript. Some sentences are barely understood. It must be reviewed and corrected. For example, line 41-44, line 82-86, and so on.

Answer: Some sentences that are barely understood have been modified. For examples, the sentences line 41-44 and line 82-86 in the original manuscript have been changed as:

In such a process, a high-solid-loading slurry consisting of raw powders, solvent, and organic binder is first required. After the slurry is casted into a mold, the organic binder creates a macromolecular network to hold the ceramic particles together. Homogeneous green bodies with required mechanical properties are therefore obtained. In the past few decades, a considerable amount of research has been done on the development of ceramic gel casting technique.” (Line 41-45 in the revised manuscript)

The flexural strength of the green bodies was measured by a universal testing machine (Z020, Zwick/Roell, German) with a crosshead speed of 1 mm/min. Samples with a size of 40 mm × 5 mm × 5 mm for the test were held in a vacuum chamber at 120 for 2 h to avoid the influence of moisture before testing.” (Line 82-85 in the revised manuscript)

Other sentences that are barely understood have also been modified in the re-submitted manuscript.

3.In the line 75, green bodies were dried at 25℃, 80℃ and 120℃. Was it done in a sequence or done separately? If was this process done separately, the green samples were tested separately?

Answer: In the line 75, the green bodies were dried at 25℃, 80℃ and 120℃ in a sequence. Thus the sentence in the line 75-76 has been changed as “To avoid cracking during drying, the green bodies were dried in a vacuum chamber at 25, 80 and 120 ℃ for 8 h in sequence.” (Line 75-76 in the revised manuscript)

4. In the line 99, authors claimed “Due to the irregular morphology and large size of TiH2 particles, it is easy to form "bridges" between TiH2 particles in the green body, while fine Al particles can fill the gap between TiHparticles and improve the density of the green body.” This sentence was not supported by evidences or relevant references. Otherwise, it shall be rewritten in a speculation manner.  

Answer: The sentence in the line 99 was not supported by evidences and relevant references. Therefore, in the revised manuscript, we decided to delete this sentence in order to avoid ambiguity.

5. In the line 108, the sentence, “There are mainly three mechanisms of dispersion, namely, steric hindrance, electrostatic repulsion and their combinations. As for non-aqueous gel-casting, steric hindrance dominates”, needs to have relevant references.

Answer: we gave a relevant reference (line 107, line 284-285) in the revised manuscript as the following:

27.       Studart, A.R.; Amstad, E.; Gauckler, L.J. Colloidal stabilization of nanoparticles in concentrated suspensions. Langmuir. 2008, 23, 1081-1090.

6. In the line 114, authors claimed “precipitation”. It seems not to be correct expression. It is better to change this expression.

Answer: “precipitation” is not a correct expression. We change the original sentence “During the preparation of TiH2/Al slurries, it was found that when the ratio of PVP to powder mass was less than 1wt.%, TiH2 particles were easy to precipitate due to their big size and density of 3.8g/cm3, which would result in non-uniform distribution of powders.” by “During the preparation of TiH2/Al slurries, it was found that when the ratio of PVP to powder mass was less than 1 wt %, TiH2 particles were easily sedimented due to their big size and density of 3.8 g/cm3, which would result in a non-uniform distribution of powders.” (Line 110-112 in the revised manuscript)

7. In the line 114, authors claimed “pseudo-plastic fluids”. Proper definition or relevant references shall be provided.

Answer: A relevant reference [28] that descripts “pseudo-plastic fluids” has been provided in the revised manuscript. (Line 145 and line 286-287 in the revised manuscript)

28.       Lin, T.R.; Lin, J.F. The elastohydrodynamic lubrication of line contacts with pseudoplastic fluids. Wear. 1990, 140, 235-249.

8. In the line 154, authors claimed “This is because that the increasing of monomer content brings more polymer networks in per unit volume, leading to the growth of strength”. This sentence was not supported by evidences or relevant references. Otherwise, it shall be rewritten in a speculation manner.

Answer: The sentence in the line 154 of the original manuscript has been rewritten in a speculation manner as following:

“This might be due to the increasing monomer content bringing more polymer networks in per unit volume, leading to increased strength. (Line 150-151 in the revised manuscript)

9. In the lines 166-170, authors claimed that bonding strength of polymer network is much stronger than TiH2 particles. In the figure 6 (f), TiH2 particles were fractured during the 3 point bend test. However, it is hardly acceptable. According to the references (J. J. Xu wt al, J. of Alloys and Compound, 436, 2007, p.82, and “Ti powder Metallurgy; Science, Technology and Application”, eds, Ma Qian and F. H. Froes, 2016, p.128, Butterworth-Heineman), yield stress of TiH2 is 774-1037 MPa, and 470-480 MPa, respectively. Under the flexural strength level of this test condition TiH2 powders would be safe. The morphology of TiHpowders in Fig. 1(a) looks facetted and the morphology which is claimed as “fractured” in Fig 6(f) could not be distinguished with pristine powders clearly. This fact shall be reconsidered carefully.

Answer: The reviewer’s comment is right. We re-observed the SEM photos and reconsidered the fact. Although some TiH2 particles could be observed from the photos, but no evidence that could prove that the TiH2 particles cracked during the bending strength tests. In order to be more scientific and rigorous, we chose to delete the following wrong explanation as following:

“It is also found from the experiment that when the monomer content is too high (for example, over 35 vol %), the slurry solidifies quickly and is not easy to control. Meanwhile, in the processes of degreasing and sintering, more polymer in the green bodies might lead to more residues, which would have adverse effects on the properties of sintered bodies. Therefore, the suitable monomer dosage determined in this study is 30 vol %.” (Line 162-167 in the revised manuscript)

10. In the line 198, authors again claimed “The fracture cleavage surface of typical TiAl intermetallics is shown in the section morphology”. The fact was observed after a mechanical testing until fracture?  If so, mechanical testing after sintering shall be discussed in the experimental and discussion.

Answer: The morphology of the TiAl intermetallic was not observed from a mechanical testing, but the surface of the sintered component. Therefore, the sentence “The fracture cleavage surface of typical TiAl intermetallics is shown in the section morphology” is a wrong explanation. In the revised manuscript, we rewrote this part as the following:

“Figure 8 shows the microstructure of the TiAl alloy sintered at 1400 ℃ under vacuum conditions for 2 h. It can be seen from Figure 8 that some open pores with sizes of several microns exist in the sintered part.” (Line 191-193 in the revised manuscript)

The manuscript has been resubmitted to the journal. We are looking forward to your positive response.

Sincerely,

Dr. Fei Li

Shanghai Key Laboratory for High Temperature Materials and Precision Forming, Shanghai Jiao Tong University, Shanghai 200240,China

Email address: lifei74@sjtu.edu.cn

Tel/Fax: +86-21-34202951

Round 2

Reviewer 2 Report

It can be seen that authors give much effort to revise this manuscript carefully per each comment. It seems to be of better state and can be considered to make a decision for the publication in this journal. However, some part of the manuscript has the English problems. It shall be carefully corrected by native speakers

Author Response

Dear reviewer,

Thank you for the valuable comments and suggestions on our manuscript. We have entrusted MDPI English Editing to improve the manuscript in English. The following is the response to your comments.

-It can be seen that authors give much effort to revise this manuscript carefully per each comment. It seems to be of better state and can be considered to make a decision for the publication in this journal. However, some part of the manuscript has the English problems. It shall be carefully corrected by native speakers.

Answer:

We sent our manuscript to the MOPI English Editing, who helped us modify the English grammar in this manuscript and sent it back to us. After careful review of the manuscript, we believe that most of the English grammar in the manuscript has been perfected.

The following is an E-mail from the MDPI English Editing:

Dear Dr. Li,
Your manuscript submitted to MDPI for English editing has been edited:
Title: Study on fabricating porous TiAl alloy via non-aqueous gel casting of a TiH2 and Al powder mixture
Length in words: 2802
English editing ID: English-8990
MDPI manuscript ID: applsci-475739
Author: Fei Li
Author email: lifei74@sjtu.edu.cn
Editing cost: 161.04 CHF
Thank you for using our editing service, here are the next steps:
1. Download the edited versions from the following link:
https://susy.mdpi.com/user/pre_english/back/8990
For Word files, edits are shown as tracked changes. For LaTeX files, we recommend software such as kdiff to see the changes made.
2. Check the changes, especially where the English editors have left comments asking for clarification about meaning.
3. Upload the approved file to the journal submission system. Don
t hesitate to contact us if you have any feedback about your edits. For tips about writing an publishing, you can follow us on LinkedIn at
https://www.linkedin.com/showcase/mdpi-english-editing-service .
Kind regards,
MDPI English Editing
MDPI
Postfach, CH-4005 Basel, Switzerland
Office: Klybeckstrasse 64, CH-4057 Basel, Switzerland
Tel. +41 61 683 77 35
Fax +41 61 302 89 18

The manuscript has been resubmitted to the journal. We are looking forward to your positive response.

Sincerely,

Dr. Fei Li

Shanghai Key Laboratory for High Temperature Materials and Precision Forming, Shanghai Jiao Tong University, Shanghai 200240China

Email address: lifei74@sjtu.edu.cn

Tel/Fax: +86-21-34202951
